# The Role of Alternative Splicing in Polyploids in Response to Abiotic Stress

**DOI:** 10.3390/ijms262010146

**Published:** 2025-10-18

**Authors:** Faiza Fatima, Mi-Jeong Yoo

**Affiliations:** Department of Biology, Clarkson University, Potsdam, NY 13699, USA; fatimaf@clarkson.edu

**Keywords:** RNA splicing regulation, stress adaptation, genome duplication, transcript isoforms, crop improvement

## Abstract

Alternative splicing (AS) is a crucial post-transcriptional regulatory mechanism that enhances transcriptomic and proteomic diversity by generating multiple mRNA isoforms from a single gene. In plants, AS plays a central role in modulating growth, development, and stress responses. We summarize the prevalence and functional roles of AS in plant development and stress adaptation, highlighting mechanisms that link AS to hormone signaling, RNA surveillance, and epigenetic regulation. Polyploid crops, with their duplicated genomes, exhibit expanded AS complexity, contributing to phenotypic plasticity, stress tolerance, and adaptive evolution. Thus, this review synthesizes current knowledge on AS in plants, with a focus on three economically important polyploid crops—*Brassica napus*, *Gossypium hirsutum*, and *Triticum aestivum*. We also discuss how subgenome interactions shape diversity in polyploids and influence trait variation. Despite significant advances enabled by high-throughput sequencing, mechanistic studies that directly link specific AS events to phenotypic outcomes remain limited. Understanding how polyploidy reprograms AS and how isoform variation contributes to stress adaptation will be critical for harnessing AS in crop improvement.

## 1. Introduction

Plants, as sessile organisms, continuously adapt to fluctuating environmental conditions that threaten their growth, development, and survival [1,2,3]. Unlike mobile organisms that can relocate to favorable environments, plants depend on complex molecular networks to perceive external cues and adjust their physiological and developmental processes accordingly [4,5]. A central component of this adaptive plasticity is post-transcriptional regulation, which dynamically modulates gene expression in response to both endogenous and exogenous stimuli [6,7,8,9,10,11,12,13,14]. Among these mechanisms, alternative splicing (AS) plays a critical role in enhancing transcriptomic complexity by generating multiple mRNA isoforms from a single gene. This diversification fine-tunes protein function and enhances plant resilience to environmental stress [15,16,17].

AS is executed through several well-defined modes that influence how pre-mRNAs are processed into mature transcripts. The five canonical AS types are intron retention (IR), exon skipping (ES), mutually exclusive exons (MXEs), alternative 5′ splice sites (A5SS), and alternative 3′ splice sites (A3SS) [18,19,20,21,22] (Figure 1).

Comparative analyses across eukaryotes have revealed clear lineage-specific patterns. In plants, IR is the dominant AS mode, accounting for approximately 30–60% of all events and closely associated with regulatory strategies such as non-sense mediated decay (NMD) coupling and stress-responsive transcriptome reprogramming [23,24]. By contrast, ES predominates in animals, contributing to tissue-specific proteome diversity and developmental plasticity [23,24,25,26]. A5SS and A3SS occur at intermediate frequencies in both, whereas MXE events are comparatively rare [23,24]. Additional sources of transcript diversity include alternative promoter (AP) usage and alternative polyadenylation (APA), which modify the 5′ and 3′ ends of transcripts. Both AP and APA events are widespread in plants, and their frequency increases under abiotic and biotic stresses [10,27,28].

Polyploidy, which results from whole-genome duplication (WGD), has been a major force in the evolution of angiosperms and strongly influences transcriptomic regulation under stress [29,30,31]. The presence of extra gene copies in polyploids facilitates subgenome divergence, relaxed selection, and expression buffering, all of which contribute to greater adaptive potential and splicing complexity [32,33]. AS further adds to this potential by increasing isoform diversity across duplicated loci. Compared to diploid plants, polyploids typically exhibit a higher frequency of IR, greater isoform divergence between homoeologs, and stronger subgenome-biased splicing, reflecting the regulatory flexibility conferred by genome duplication [5,33]. Evidence shows that AS divergence in polyploids is frequently tissue-specific and stress-responsive, providing an additional layer of post-polyploidization gene regulation [30,34,35].

AS is common across plant species but varies with genetic background, developmental stage, and environmental conditions [30,36,37]. While much work has characterized AS in *Arabidopsis thaliana* [38], *Oryza sativa* [39,40], and *Zea mays* [40], more recent studies have focused on polyploid crops such as *Brassica napus* [41], *Gossypium hirsutum* [42], and *Triticum aestivum* [43]. These crops are both economically important and genetically complex, making them valuable systems for investigating how AS influences stress adaptation and productivity [20,25].

This review synthesizes current knowledge of AS regulation in plants, with a focus on the polyploid crops *B. napus*, *G. hirsutum*, and *T. aestivum*. It summarizes recent insights into their AS landscapes, functional consequences, and evolutionary significance. By highlighting key advances and knowledge gaps, the review provides perspectives on how AS can be harnessed for crop improvement strategies.

## 2. Prevalence and Variability of AS Across Plant Species

AS is a crucial post-transcriptional mechanism that increases transcriptomic and proteomic diversity in plants. Although conserved across plant lineages, its prevalence and regulatory patterns differ markedly among species, tissues, developmental stages, and ploidy levels.

Comparative transcriptome studies have shown that 40–70% of multi-exon genes undergo AS in many model species such as *A. thaliana*, *O. sativa*, and *Z. mays* [44,45,46]. Variation in these estimates largely reflects differences in gene architecture, regulatory motifs, sequencing technologies, and read depth [47,48,49].

Tissue-specific regulation of AS enables plants to fine-tune gene function across diverse organs and developmental contexts. For example, in *Rosa roxburghii,* RNA-seq analysis across eight tissues revealed 8586 AS genes and over 49,000 AS events, with hundreds of isoforms restricted to leaves, flowers, and fruits [50]. Many of these tissue-specific AS genes (e.g., *4CL*, *DFR*, *ANR*, *MYB*, and *bHLH*) were directly associated with flavonoid synthesis, underscoring the link between AS isoform diversity and organ-specific metabolism. Similarly, in *Nicotiana tabacum,* a transcriptome atlas covering 13 tissues identified 4585 tissue-specific transcripts, most of which arose from AS [51]. Trichomes, flowers, and roots exhibited particularly high numbers of unique isoforms, many linked to secondary metabolism and stress responses, highlighting AS as a key driver of tissue-specialized functions. At the single-gene level, the study of *Musa acuminata* provides an illustrative example: AS of *MaMYB16L* generates two isoforms with opposing functions—the full-length variant represses starch-degradation genes and delays ripening, while the truncated isoform is preferentially expressed in fruit tissue, promoting softening and metabolic transition [52].

AS frequency and isoform distribution shift dynamically throughout development. In a cultivated strawberry (*Fragaria* × *ananassa*), long-read sequencing revealed over 20,000 AS isoforms, highlighting the transcriptomic complexity of this octoploid genome [53]. More recently, Nanopore-based profiling across four fruit stages identified hundreds of isoform switches affecting hormone signaling and metabolism, underscoring developmental stage-specific regulation of AS [54]. These events were frequently associated with protein domain gain or loss and hormone signaling pathways, underscoring developmental stage-specific regulation. A similar developmental modulation of AS has been described in *Brassica juncea,* where the MADS-box gene *BjuAGL18-1* produces two antagonistic isoforms. The full-length transcript (*BjuAGL18-1L*) represses flowering by forming complexes with AFR2–HDA9 and AGL15 to inhibit FUL and FT, while the truncated isoform (*BjuAGL18-1S*) counteracts this repression and promotes early flowering [55]. Their expression patterns are strongly influenced by environmental conditions, with *BjuAGL18-1L* induced under short days and *BjuAGL18-1S* favored under long days and higher temperatures, providing a clear example of how AS integrates developmental stage transitions with external cues to fine-tune flowering.

Polyploidy significantly increases AS abundance and complexity because genome duplication and functional diversification of homoeologous gene copies provide additional regulatory flexibility [56,57]. This increased complexity arises from several interacting mechanisms: relaxed purifying selection on duplicated genes allows splice-site mutations to accumulate [56]; divergence in promoter and chromatin landscapes between subgenomes alters co-transcriptional splicing decisions; and differential expression of spliceosomal components across subgenomes further amplifies isoform diversity [57]. Together, these processes create an expanded regulatory network that enables polyploids to fine-tune gene expression under stress [56,57]. For example, He et al. [5] reconstructed full-length transcriptomes of interspecific *Oryza* hybrids with different ploidy levels, revealing that tetraploid and hexaploid hybrids exhibited higher AS diversity than their diploid counterparts. These differences were particularly evident in stress-response and developmental genes, highlighting the role of genome duplication in expanding regulatory capacity through AS [5]. A similar phenomenon was observed in *B. napus*, in which the number of isoforms and the number of genes involved in AS increased compared to its diploid progenitors [58]. In addition, in *B. napus*, subgenome-dominant expression and AS have been shown to jointly shape the immune response to *Sclerotinia* infection, with the C-subgenome contributing more to both expression and splicing regulation [33]. Likewise, in coffee, genome-wide analysis of the AP2/ERF transcription factor family showed that allotetraploid *Coffea arabica* contains nearly twice as many genes as its diploid progenitors *C. canephora* and *C. eugenioides*, largely due to post-polyploid tandem and segmental duplications [59]. Subgenome-specific biases were evident, with canephora-derived chromosomes enriched in disease-resistance genes and eugenioides-derived chromosomes retaining more temperature-tolerance genes, illustrating how polyploidy diversifies regulatory repertoires under stress.

These findings show that AS is a pervasive yet highly variable regulatory process. Its prevalence is shaped not only by intrinsic genomic features, but also by tissue identity, developmental stage, and ploidy level, each of which introduces additional layers of regulatory complexity.

## 3. The Role of AS in Plant Growth and Development

AS influences a wide range of developmental processes, but its functional roles have been most clearly demonstrated in flowering regulation, germination, hormone transport, and shoot architecture [10,27,60]. This section summarizes representative cases in which specific splice isoforms, splicing factors, or splicing-related mutations have been mechanistically linked to developmental outcomes, providing clear evidence of AS function during plant growth.

Numerous studies have demonstrated that AS regulates flowering in many plant species (reviewed in [18]), by generating distinct isoforms of flowering time genes such as *FLOWERING LOCUS T* (*FT*) [61] and *FLOWERING LOCUS C* (*FLC*) [62], as well as transcription factors including the bZIP transcription factor FD [63], MADS-box transcription factor *FLOWERING LOCUS M* (*FM*) [64,65], and *CONSTANS* (*CO*) [66,67]. For example, splicing factor 30 (SR30) regulates flowering primarily through *FLC*, while its own function is self-regulated by AS [68]. The balance between functional and NMD-targeted isoforms fine-tune SR30 abundance of SR30, thereby modulating flowering time.

In *Arabidopsis*, the spliceosomal component SmEb modulates abscisic acid (ABA) sensitivity during germination by maintaining the proper splicing balance of HAB1, a negative regulator of ABA signaling. Loss of SmEb disrupts this balance, resulting in increased ABA sensitivity and delayed cotyledon greening [69]. Similarly, in barley, the cap-binding complex (CBC), composed of CBP20 and CBP80, regulates ABA-responsive AS during germination. Double mutants lacking CBC subunits exhibit altered splicing of ABA-related genes and reduced ABA sensitivity, suggesting that the CBC functions upstream of ABA signaling through AS control [70].

Auxin-mediated development is also regulated through AS of *PIN7* in *Arabidopsis*. Two isoforms, *PIN7a* and *PIN7b*, have opposite effects on auxin transport and developmental processes such as apical hook formation and hypocotyl bending. Their co-expression and physical interaction indicate that a precise isoform balance is essential for proper tropic responses [71].

In *Gossypium arboreum*, a single-nucleotide mutation in the *TFL1* gene alters the canonical 5′ splice site, producing a truncated isoform that disrupts shoot apical meristem maintenance. This mutation leads to determinate growth and terminal flower formation. Transcriptomic analysis of the mutant revealed altered expression of auxin and sugar transporters, suggesting that *TFL1*-mediated AS integrates hormonal and metabolic signals during shoot development [72].

In polyploid species, several studies have investigated the roles of AS in the development of cotton and wheat. As a major textile crop, AS profiles of cotton were examined during fiber development in *G. hirsutum* [73] and *G. barbadense* [74]. Earlier work demonstrated that AS of MADS-box transcription factors in G. hirsutum fiber cells generates multiple isoforms with distinct expression patterns and possible regulatory functions [75], supporting the hypothesis that AS contributes to tissue-specific differentiation and secondary cell wall biosynthesis in cotton fibers.

In wheat, several recent studies have revealed how alternative splicing (AS) fine-tunes developmental processes and yield formation. The *TaPP2C-a5* gene produces two splice isoforms, *TaPP2C-a5L* and *TaPP2C-a5S*, with opposing effects on abscisic acid (ABA) signaling and seed dormancy: the Triticeae-specific short isoform enhances dormancy and ABA sensitivity, whereas the long isoform promotes germination [76]. An intron-located single-nucleotide variation in *TaGS5-3D* alters splicing efficiency, leading to intron-retained transcripts that increase grain size and yield [77]. Similarly, *TaNAK1* generates two isoforms, *TaNAK1.1* and *TaNAK1.2*, which exert opposite regulatory effects on flowering and plant architecture—*TaNAK1.2* accelerates flowering and increases plant height and seed yield, whereas *TaNAK1.1* delays flowering and restricts growth [78]. Another AP2-like gene, *TaTOE1-B1*, produces three splice variants, of which *TaTOE1-B1-3* acts as a floral inhibitor by repressing FT and SOC1 expression, thus delaying the transition to flowering [79]. Together, these studies demonstrate that AS functions as a dynamic post-transcriptional mechanism that coordinates hormone signaling, flowering, and yield traits in polyploid wheat.

Collectively, these examples show that AS can modulate signaling pathways (e.g., ABA and auxin transport) or generate variable isoforms that drive distinct developmental outcomes, highlighting its critical role in shaping plant growth and development.

## 4. Abiotic Stress-Induced Alternative Splicing Variants

Abiotic stresses such as drought, salinity, and extreme temperatures profoundly alter plant transcriptomes. AS acts as a rapid and flexible regulatory layer, frequently targeting transcription factors, RNA-binding proteins, and signaling components to modulate hormone pathways, ion transport, and metabolic responses. Many isoform shifts occur independently of overall transcript abundance, underscoring AS as a distinct mechanism of stress adaptation. This section highlights how drought, salinity, and temperature reshape splicing landscapes and influence plant resilience in model and major crop species such as *Arabidopsis*, maize, rice, and so on. These examples illustrate general regulatory principles that are also conserved or amplified in the three focal polyploid crops described in Section 5.

### 4.1. Drought Stress

Drought is a major abiotic stress that limits plant growth and productivity, and AS has emerged as a critical mechanism for enhancing drought tolerance by modulating stress-responsive gene expression [6,80].

In maize, the clade B phosphatase gene *ZmPP2C26* undergoes alternative first exon splicing to produce isoforms with distinct activities, both of which negatively regulate drought tolerance [81]. Mutant lines lacking *ZmPP2C26* exhibit improved photosynthesis and water retention, linking AS directly to drought sensitivity. Epigenetic regulation also influences splicing outcomes, as DNA-methylation changes near splice junctions affect drought-induced AS in genes related to photosynthesis and stress signaling [82].

Similar AS-mediated regulation has been observed in soybean, where thousands of drought-responsive events involve transcription factors (e.g., MYB, NAC, bZIP) and splicing factors (e.g., SR45a, U2AF), many of which display stage-specific or inverse expression relative to their parent genes [83].

In barley, ABA-dependent splice variants of *HvABI5* and serine/arginine-rich (SR) proteins modulate hormone-responsive pathways, while an isoform of *HvLHCA4.2-S* enhances ROS scavenging and stomatal closure, improving drought tolerance in transgenic plants [84,85].

Rice studies have highlighted a role for AS in drought memory, identifying nearly 2000 memory-associated AS events, enriched in RNA-binding proteins and transcription factors, that contribute to stress imprinting and recall [86]. Also, *OsbHLH59* produces two isoforms, *OsbHLH59.1* and *OsbHLH59.2*, the latter of which is induced by ABA and enhances drought tolerance [87].

These studies demonstrate that drought-induced AS frequently targets transcription factors, ABA signaling components, and stress-protective genes. However, functional validation remains limited to a few genes and species, underscoring the need for broader mechanistic studies in crops.

### 4.2. Salinity Stress

Salinity stress leads to large-scale transcriptomic reprogramming, and AS is increasingly recognized as an important regulator of salt-tolerance pathways [11,88]. Several studies have identified specific splice variants, splicing factors, and natural splice site polymorphisms that contribute to salt stress responses.

In *Arabidopsis*, *Salt-Responsive Alternatively Spliced gene 1* (*SRAS1*), which encodes a RING-type E3 ubiquitin ligase, produces two splice variants, *SRAS1.1* and *SRAS1.2*, that display contrasting responses to salt stress [89]. Under salt stress conditions, the alternative splicing event favors increased accumulation of *SRAS1.1* and a reduced level of *SRAS1.2*.

In *Populus trichocarpa*, the SR-rich factor PtSC27 is strongly induced under saline conditions [90]. Overexpression in *Arabidopsis* altered splicing patterns of ABA-responsive genes, suggesting a role in fine-tuning hormone-regulated isoforms during salt adaptation.

Long-read sequencing in rice identified extensive early AS responses to salinity, dominated by IR. Stress-responsive transcription factors, ion transporters, and SR proteins (e.g., RS33, RS2Z36) displayed genotype-specific AS events, implying a feedback loop that modulates splicing efficiency in tolerant genotypes [91]. Complementing this, Genome-wide association studies (GWAS)–AS integration revealed 764 genotype-specific splicing events, including variants in *OsRAD23* and *OsNUC1* linked to Na^+^ accumulation and salt-tolerance diversity [92]. Moreover, in *Gossypium* species, 47 sodium/hydrogen antiporter (*NHX*) genes with long introns were identified, indicating high potential for AS-mediated diversification of ion transporters [93].

Polyploidization may further expand AS diversity under salt stress. Evidence from legumes shows that autopolyploidization reprograms post-transcriptional regulation [94], while polyploid citrus exhibits epigenetic remodeling of ethylene-biosynthesis genes, potentially enhancing stress resilience through expanded isoform variability [95]. Salinity-induced AS predominantly affects transcription factors, ABA-responsive regulators, and ion-transport genes. Polyploid species may further exploit their increased gene copy number to expand AS diversity and enhance salt tolerance.

### 4.3. Temperature Stress

Temperature extremes strongly influence plant development and productivity, and numerous studies have revealed that AS is a key regulatory layer in both cold and heat stress responses. These studies particularly implicate transcription factors, heat shock proteins, and splicing regulators as major AS targets.

In *A. thaliana*, the spliceosomal component SmEb is essential for proper splicing of *HAB1* and other stress-related genes under chilling stress. SmEb-deficient mutants show chlorophyll degradation, abnormal development, and heightened cold sensitivity, linking splicing fidelity to stress adaptation [96]. A related study revealed that CBF transcription factors promote condensation of splicing regulators (RS40, RSZ32) into nuclear speckles, enhancing cold-responsive AS; mutations in these components impair isoform switching and reduce cold tolerance [97].

Cold-induced AS has also been documented in poplar (*P. trichocarpa*, *P. ussuriensis*), where >14,000 events—mainly IR—affect numerous transcription factors (PtrNAC72, PtrCBF1, PtrDREB2A) and splicing regulators, indicating feedback control of the splicing machinery itself [98].

Heat stress likewise triggers AS of transcription factors and chaperone regulators. In maize, *ZmHSF12*, heat shock transcription factor, encodes two alternative spliced transcripts, *ZmHSF12-1* and *ZmHSF12-2*, which regulate the balance of plant growth and heat tolerance [99]. In *Arabidopsis*, *RDM16*, a pre-mRNA splicing factor, produces two isoforms—*RDL* (long) and *RDS* (short) [100]. The *RDS* facilitates the condensation of *RDL*, synergistically strengthening their function in heat tolerance. Heat stress also modulates spliceosome activity, for example, the dephosphorylation of spliceosome subunits by PP2A B’η modulates the splicing pattern, leading to the production of different protein isoforms in response to heat stress [101].

Other mechanisms include exitron splicing—cryptic introns located within annotated exons of MBD4L in *Arabidopsis*, which alters nuclear localization of the resulting isoforms [102]. In waxy maize, salicylic acid pretreatment reduced the frequency of heat-induced AS events and restored the normal expression patterns of key heat-responsive genes, thereby enhancing thermotolerance [103].

Overall, both cold and heat stress induce widespread AS changes, particularly in transcription factors, splicing regulators, and heat-shock proteins. Many of these genes form feedback loops in which the splicing machinery itself undergoes stress responsive AS, amplifying transcriptome flexibility.

### 4.4. Molecular Pathways Linked to AS Under Stress

Environmental stresses such as drought, salinity, and high temperature extensively reprogram AS patterns at the genome-wide level. These changes are not random but are regulated through interconnected pathways involving splicing factors, hormone signaling, RNA surveillance, and chromatin dynamics (Figure 2A–D). Although the core spliceosome (U1, U2, U4/U6, and U5 snRNPs) remains intact, stress conditions influence splice-site selection by modulating the activity, expression, and isoform diversity of associated regulators—particularly SR proteins and heterogeneous nuclear ribonucleoproteins (hnRNPs) (Figure 2A) [104,105,106].

Several SR proteins, such as SR30, SR45, and RSZp22, are themselves regulated by AS under stress. For example, heat stress in *Arabidopsis* triggers AS of SR34b and SF1, which subsequently modulates the splicing of heat-responsive genes including *HSFA2* and *HSP21* [12,107]. This feedback regulation forms a dynamic loop in which splicing factors undergo AS to fine-tune downstream gene expression, enabling plants to rapidly adapt to temperature fluctuations.

Hormone signaling is another major regulator of stress-responsive AS. ABA pathway is strongly influenced by the AS of key components such as PYL receptors, SnRK2 kinases, and downstream transcription factors. These events often generate truncated or nonfunctional isoforms that alter ABA sensitivity (Figure 2B), leading to either enhanced or reduced stress responses depending on the environmental context [88,108,109].

AS is also tightly linked to NMD, which degrades transcripts containing premature stop codons. Many stress-induced isoforms are predicted NMD targets; however, some escape degradation by remaining in the nucleus, potentially acting as reservoirs for rapid translation during recovery (Figure 2C) [22,60,110]. This coupling between AS and NMD provides an additional layer of gene expression control in fluctuating environments.

Epigenetic modifications and chromatin remodeling also influence AS under stress. Changes in DNA methylation, histone modifications, and RNA polymerase II elongation rate can affect co-transcriptional splicing decisions. Stress-induced chromatin state changes have been shown to shift exon–intron boundaries and increase the frequency of IR or ES events (Figure 2D) [60,111].

Importantly, transcriptome analyses reveal that AS and gene-expression levels often act independently, affecting distinct gene sets. This uncoupling allows plants to produce functionally distinct isoforms without changing overall transcript abundance, offering a rapid means of stress adaptation [27,80].

Finally, advances in long-read sequencing (e.g., Iso-Seq) and splicing-aware CRISPR technologies are accelerating the discovery and functional validation of stress-responsive isoforms. These tools are expected to facilitate the targeted manipulation of AS pathways to develop stress-resilient crop varieties [6].

## 5. Alternative Splicing in Polyploids

This section provides a comparative synthesis of AS patterns across the three focal polyploid crops, including predominant event types, representative genes responsive to biotic or abiotic stresses, and functional outcomes. As polyploids often exhibit enhanced stress modulation compared to their diploid progenitors, they serve as ideal systems for investigating how duplicated genomes interact with AS to expand regulatory capacity under stress. In the following subsections, we summarize recent advances in *B. napus*, *G. hirsutum*, and *T. aestivum*—which are widely studied due to their economic importance, complex genome organization, and well-developed genomic resources.

### 5.1. Brassica napus

*Brassica napus* is an economically important allotetraploid oilseed crop (AACC) formed by hybridization between *B. rapa* (A-genome) and *B. oleracea* (C-genome) approximately 7500 years ago [112]. AS is a key post-transcriptional mechanism that contributes to transcriptomic diversity, stress adaptation, and phenotypic variation in this species.

High-resolution isoform studies using PacBio Iso-Seq and Illumina RNA-seq (Beijing Genomics Institute, Wuhan, China) identified 147,698 isoforms and 222,000 AS events, of which 69% were tissue-specific [113]. Approximately 31,500 events altered open-reading frames or protein domains, indicating potential functional diversification.

Comparative analyses of natural and resynthesized *B. napus* lines showed that 26–30% of duplicated genes exhibit AS divergence relative to their diploid parents, with many AS losses or shifts converging across independently resynthesized lines [114]. Notably, natural *B. napus* exhibited more extensive AS responses than resynthesized lines, reinforcing the idea that genomic interactions and subgenome dominance—where resynthesized B. napus plants showed a greater subgenome bias under infection, with the C-subgenome contributing disproportionately to the defense response—co-evolve with AS regulation [33]. Complementing these findings, Sun et al. (2024) directly compared *B. napus* with its diploid progenitors (*B. rapa* and *B. oleracea*) across four tissues (stems, leaves, flowers, and siliques) [115]. They showed that *B. napus* harbors a greater number of AS events than either progenitor during allopolyploidization. These splicing changes were associated with DNA methylation dynamics and *cis*/*trans* regulatory divergence, indicating that polyploidization expands isoform diversity while reshaping epigenetic control. Although many duplicated genes remain conserved, a substantial fraction undergoes neofunctionalization or specialization, highlighting the interplay between AS divergence, epigenetic remodeling, and subgenome interactions [115].

Extending these observations to environmental regulation, long-read sequencing revealed that temperature stress in *B. napus* alters both the frequency and diversity of AS events. Heat stress increased the number of AS events and the proportion of isoforms predicted to undergo NMD, indicating heightened post-transcriptional regulation [116]. Conversely, cold stress reduced isoform diversity per gene. The study also revealed subgenome-biased splicing, with C_T_-homeologs producing more isoforms than A_T_-homeologs across conditions [116].

AS is strongly associated with both biotic and abiotic stress responses. IR is the predominant AS type and is especially enriched under *Sclerotinia sclerotiorum* [33,117] and *Leptosphaeria maculans* [118] infections, where it affects genes encoding splicing factors, transcriptional regulators, and defense-related proteins. Similarly, abiotic stresses such as boron deficiency [119], heat, cold, drought, salinity, and dehydration [120,121] induce AS in genes involved in RNA binding, signal transduction, and stress-responsive pathways. In general, both biotic and abiotic stress increase the diversity of mRNA transcripts by inducing more AS events. Moreover, functional studies in *B. napus* have demonstrated that specific isoforms contribute to stress adaptation, linking AS variation to enhanced immune responses and abiotic stress tolerance [33,117,118,120]. A recent study revealed that *BnABF4L*, an ABA pathway regulator, undergoes A3SS-mediated alternative splicing to produce three isoforms (V1–V3) [41]. Stress conditions selectively repress V1 translation but favor V2 and V3, which together modulate stress responses.

Beyond stress responses, AS contributes to developmental variation. Splicing differences in flavonoid-biosynthesis genes are linked to seed-coat color [122], and modifications in spliceosome components affect secondary dormancy [123].

Subgenome-level divergence in AS has been widely observed. Under cold, heat, and drought stress, the A-subgenome tends to dominate gene expression, whereas the C-subgenome exhibits greater AS activity across more than 27,000 gene pairs [121]. AS frequency was negatively correlated with gene expression, and cold stress triggered the most extensive splicing changes, with cold–drought overlap being greater than cold–heat overlap [121]. A transcriptome-wide analysis of multiple abiotic stress (cold, dehydration, salt, and ABA) responses in *B. napus* identified 357 differentially alternatively spliced (DAS) genes, predominantly IR events. Weighted gene co-expression network analysis (WGCNA) revealed 23 modules associated with stress responses, highlighting hub genes such as *RBP45C* (RNA-binding protein regulating AS), *LHY* (circadian clock regulator), *MYB59* (MYB transcription factor linked to cell cycle and stress responses), *SCL30A* (GRAS-family splicing factor), *SR40* (serine/arginine-rich splicing regulator), *MAJ23.10* (putative RNA-binding protein), and DWF4 (brassinosteroid biosynthesis enzyme) [120].

Overall, studies in *B. napus* consistently demonstrate that AS is highly dynamic, stress-responsive, and shaped by subgenome interactions. The interplay of expression dominance and AS divergence highlights the role of polyploidy in expanding transcriptome complexity and adaptive potential.

### 5.2. Gossypium hirsutum

The genus *Gossypium* is a well-established model for studying polyploid genome evolution. Allotetraploid cotton species, including *G. hirsutum* (AD_1_), originated 1–2 million years ago from a hybridization event between an A-genome-like African diploid and a D-genome-like American diploid [124,125]. Since polyploidization, *G. hirsutum* has undergone asymmetric subgenome domestication and artificial selection for more than 4000–5000 years [126,127]. Survey-based analyses have shown that approximately 27–30% of multi-exon genes in allotetraploid cotton undergo AS, with IR as the most frequent event type, establishing AS as a pervasive regulatory layer in this crop [128].

Comparative genome sequencing of diploid and tetraploid cotton species has revealed asymmetric evolution and biased gene expression between the A_t_ (A-subgenome) and D_t_ (D-subgenome) components [129,130,131,132,133]. However, until recently, the lack of well-annotated transcript isoform datasets limited detailed analysis of AS divergence between homoeologous genes [132,133].

Abiotic stress studies further reveal that splicing plasticity is integral to cotton adaptation. Integrated transcriptomic and proteomic profiling demonstrated that salt tolerance is mediated through AS-dependent regulation of ion transport and protective pathways [134]. More recently, combined transcriptomic and proteomic analysis under salt and drought stress showed widespread splicing reprogramming dominated by IR, with several isoforms validated as regulators of stress response pathways [135].

Biotic interactions also induce extensive isoform switching. Herbivory and pest challenge triggered differential splicing of defense-related genes, including transcription factors and hormone-responsive regulators, highlighting AS as a key component of cotton defense networks [136]. AS additionally influences developmental and metabolic traits. Isoform switching in gossypol biosynthesis genes contributes to gland morphogenesis, linking AS to the regulation of secondary metabolism and tissue specialization in cotton [137].

More recent work has emphasized the stage- and tissue-specific nature of AS during fiber development. Approximately 23.3% of multi-exon genes undergo AS during this process, with events being more common in ovules and early fiber initiation stages than in mature fibers [73]. Interestingly, IR was not the dominant AS form during these stages, suggesting that splicing regulation is developmentally modulated. Gene-body methylation and splicing factor activity were implicated as regulators of these patterns, revealing an epigenetic component in cotton fiber transcriptome regulation [73].

Despite these advances, direct comparisons of AS landscapes between *G. hirsutum* and its diploid progenitors remain scarce, with most insights coming from analyses of subgenome-biased splicing within the allotetraploid itself. These studies demonstrate that AS in *G. hirsutum* is a critical layer of transcriptome regulation shaped by polyploid genome architecture, developmental context, and epigenetic control. Isoform divergence between homeologs and spatiotemporal AS patterns underscore the functional complexity introduced by polyploidy in cotton.

### 5.3. Triticum aestivum

Bread wheat (*T. aestivum*) is an allohexaploid species (2*n* = 6*x* = 42) that originated approximately 8000 years ago through hybridization between the cultivated allotetraploid *T. turgidum* (AABB) and the diploid goatgrass *Aegilops tauschii* (DD), resulting in the AABBDD genome configuration [74,138]. This polyploidization has contributed to the structural complexity and plasticity of the wheat genome, shaping its adaptability under both natural and artificial selection [139].

AS is a major post-transcriptional regulatory mechanism in wheat, especially under abiotic stress conditions. A transcriptome analysis of Najran wheat under 200 mM NaCl revealed that 22.5% of root-expressed genes and 23.1% of shoot-expressed genes underwent AS [43]. Unlike *B. napus* and *G. hirsutum*, where IR predominates, A3SS was most common in wheat, while MXE was least frequent. Salt stress also induced tissue-specific regulation, with 82% of AS events differing between roots and shoots. Functional annotation showed that root AS events were enriched in cytoskeleton-related processes, whereas shoot AS events were associated with metabolism and signal transduction [43]. Similar patterns were observed in comparative analyses of salt-tolerant and salt-sensitive cultivars, where more than 9000 differentially expressed genes and extensive AS reprogramming were detected. Interestingly, AS isoforms of SR protein genes were more prevalent than changes in their overall expression, underscoring that splicing regulation plays a central role in salt adaptation [140].

Heat stress also triggers AS of transcription factors and chaperone regulators. *TaHsfA2-7*, heat shock transcription factor, produces a truncated isoform (*TaHsfA2-7-AS*) that accumulates in the nucleus and enhances heat tolerance during flowering and grain-filling, with *TaHsfA1* directly regulating both isoforms [141]. Another heat shock transcription factor, *TaHSFA6e*, generates two isoforms under heat stress, with the extended *TaHSFA6e-III* more strongly activating *TaHSP70* genes and promoting stress-granule disassembly during recovery [142].

Consistent with these findings, a genome-wide AS analysis comparing the salt-sensitive cultivar Chinese Spring and the salt-tolerant cultivar Qing Mai 6 identified 11,141 genes with salt-responsive AS events [143]. Chinese Spring exhibited more events (21,203) than Qing Mai 6 (19,742), suggesting salt stress induces more AS events. AS events were distributed across the A, B, and D subgenomes (7235; 8143; 7407 events, respectively), with AS activity peaking 48 h after stress onset.

AS is also stress-specific: drought (DS), heat (HS), and combined heat–drought (HD) treatments induced AS in 200, 3576, and 4056 genes, respectively. The B-subgenome showed the highest AS activity across all conditions. Notably, only 12% of DS-responsive AS genes overlapped with differentially expressed genes, compared to ~40% under HS and HD, indicating that AS plays a particularly important role in post-transcriptional regulation under heat and combined stress [144]. Studies of thermosensitive genic male-sterile (TGMS) lines further illustrate this connection, where nearly 8000 differentials AS events were identified in developing pollen. These events disrupted modules related to cytoskeleton dynamics, calcium transport, and vesicle trafficking, directly linking AS to fertility regulation under stress [145].

Beyond stress responses, AS contributes to wheat development and polyploid genome evolution. Isoform profiling during embryogenesis revealed extensive splicing dynamics, with IR most frequent and homoeolog-specific divergence evident across subgenomes, indicating that AS underlies both developmental regulation and evolutionary diversification [32]. Likewise, analyses of newly formed synthetic hexaploid wheat demonstrated that homoeologous exchanges reshape both gene expression and AS landscapes, showing how structural genomic changes introduce new layers of splicing variation [146]. Comparative analyses of wild and domesticated wheat further revealed that both domestication and polyploidization reduced AS frequency, largely through loss of IR events and downregulation of splicing regulators [147]. This observation contrasts with *B. napus*, in which more AS events were detected than in either diploid progenitor [33]. The difference may stem from contrasting experimental contexts, namely wheat seedlings under non-stress conditions versus mature *Brassica* leaves experiencing biotic stress. Many homoeolog triplets exhibited partitioned AS patterns, highlighting the combined role of subgenome interaction and selection in shaping the wheat AS landscape [147].

These studies reveal that AS in hexaploid wheat is both stress- and subgenome-dependent, with distinct isoform profiles across tissues, genotypes, and stress types. The dynamic nature of AS highlights its potential for isoform-level breeding strategies to enhance wheat stress tolerance.

Across the three polyploid crops, AS shows both conserved and lineage-specific patterns. Heat and drought are the most extensively studied stresses, with IR prevailing in *B. napus* and *G. hirsutum*, whereas *T. aestivum* exhibits more ES [145] and A3SS [43]. Despite these differences, stress-induced splicing of transcription factors, RNA-binding proteins, and hormone-responsive genes is a unifying theme. Cold- and salinity-related AS remain less explored, particularly in *T. aestivum* and *G. hirsutum*, indicating future research needs. In addition, given that most current reports are descriptive, detailed functional studies of genes producing AS isoforms are needed to determine how AS regulates gene expression and contributes to stress resilience in polyploids. A conceptual summary of splicing divergence following polyploidization across these three species is illustrated in Figure 3, emphasizing the deviation between expected additive expression and the observed complexity resulting from subgenome interactions and stress-responsive regulation.

### 5.4. Other Polyploid Species

Beyond these major crops, AS profiles have also been characterized in other polyploid systems such as soybean [148], *Camelina sativa* [149], *Camellia oleifera* [150], peanut [151], and cotton relatives (*G. barbadense*) [74], all showing a predominance of IR. However, most of these studies have primarily cataloged AS events rather than elucidating their regulatory significance. Further research is needed to determine whether the mechanisms summarized here—subgenome-biased splicing, stress-responsive isoform switching, and epigenetic modulation—represent conserved features across diverse polyploid lineages.

## 6. Future Directions and Implications

Research over the past decade has greatly expanded our understanding of AS as a key post-transcriptional regulatory layer in plants. High-throughput short- and long-read sequencing studies have revealed that polyploid crops such as *B. napus*, *G. hirsutum*, and *T. aestivum* exhibit extensive AS diversity shaped by subgenome interactions, epigenetic regulation, and environmental signals [74,113,121]. Despite these advances, most findings in those three species remain descriptive. For instance, while heat- and drought-responsive isoforms have been cataloged in *B. napus* and *T. aestivum*, and tissue-specific AS characterized in cotton fibers, their direct physiological or yield-related functions are still unclear.

Although transcriptomic surveys consistently reveal widespread AS in plants, the extent to which it translates into functional protein diversity is unresolved. AS isoforms can alter protein properties, including tissue-specific expression, subcellular localization, domain composition, or interaction partners [152,153]. For instance, splice variants may localize to distinct organelles (e.g., mitochondria vs. chloroplasts), act as dominant-negative regulators of transcription factors, or engage in cooperative loops that fine-tune signaling pathways [154,155]. Beyond localization, AS can modulate protein functionality by altering catalytic or binding domains, post-translational modification sites, or protein stability [153,154,155]. Isoform-specific differences in phosphorylation, ubiquitination, or acetylation can redirect signaling fluxes or determine degradation rates, linking AS directly to proteostasis and stress adaptation [154]. Recent proteomic and interactomic analyses have also shown that AS frequently rewires protein–protein interaction networks, revealing isoform-dependent partners that contribute to tissue-specific and stress-responsive functions [153,154]. These mechanisms underscore that AS not only increases transcriptome complexity but also provides a versatile toolkit for expanding proteomic outputs, with consequences for stress adaptation, development, and evolutionary innovation [155,156].

Future research should prioritize experimental validation of isoform functions, using CRISPR-based splicing modulation, isoform-specific overexpression, and gene-editing approaches to link AS events to cold, heat, drought, and salinity resilience as well as yield stability. Integrating AS datasets with epigenomic, proteomic, and metabolic profiles will be essential to uncover how AS interacts with other regulatory layers. Moreover, the development of isoform-aware molecular markers could enable breeders to incorporate favorable splice variants into breeding programs, complementing traditional gene-based selection strategies [80].

Recent work also links AS regulation to plant immune responses. The fungal pathogen *Exserohilum turcicum* secretes the effector EtEC81, which interacts with the maize splicing regulator ZmEIP1 to reprogram defense-related splicing and enhance immune signaling [157]. Such pathogen-driven manipulation of host AS demonstrates that splicing regulation extends beyond abiotic stress adaptation to encompass biotic stress and host–pathogen co-evolution. To translate these insights into breeding practice, future programs could integrate isoform-specific quantitative trait loci (iso-QTL) mapping and develop high-throughput assays to detect favorable splice variants in elite germplasm. In parallel, CRISPR/Cas-based genome and splice-site editing can be used to fine-tune intron–exon architecture or modulate splicing factors, creating cultivars with optimized stress-responsive isoform profiles.

As climate change intensifies abiotic and biotic stresses, harnessing AS in key polyploid crops, such as *B. napus*, *G. hirsutum*, and *T. aestivum*—offers a promising but under-explored path for improving resilience and productivity. Their duplicated genomes provide natural flexibility for fine-tuning splicing patterns through breeding and genome editing, making them ideal models for developing AS-informed improvement strategies. Future studies should aim to link transcript isoform diversity to adaptive traits and productivity. Strengthening collaborations between molecular biologists, breeders, and computational scientists will be critical to translating AS knowledge into practical applications for sustainable agriculture.

## Figures and Tables

**Figure 1 ijms-26-10146-f001:**
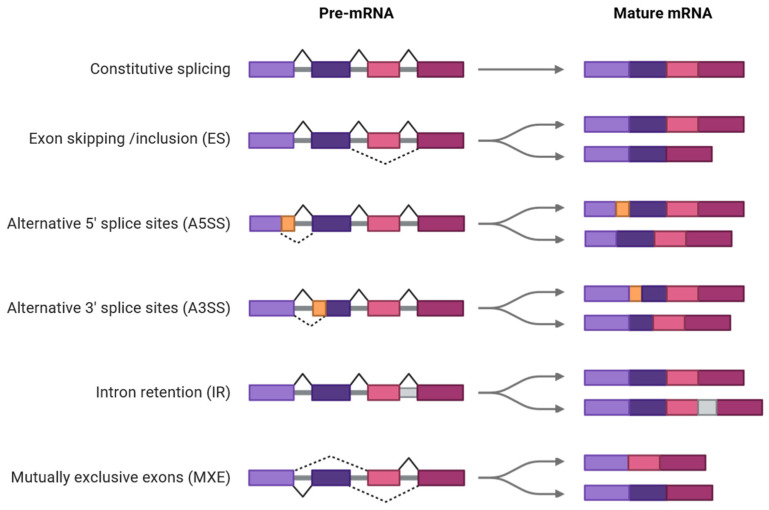
Canonical modes of AS in plants. Solid and dotted lines in pre-mRNA indicate constitutive splicing and AS events, respectively. Diagrams were created in BioRender. Faiza Fatima. (2025). https://app.biorender.com/illustrations/68c05363ab47b2405e081788 (accessed on 12 September 2025).

**Figure 2 ijms-26-10146-f002:**
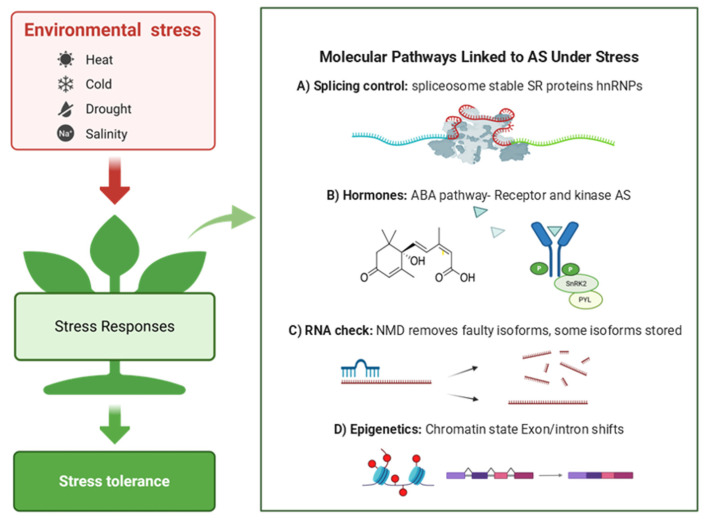
Conceptual model of stress responsive AS in plants. Environmental stressors (heat, cold, drought, salinity) are perceived and transduced to elicit cellular responses that promote stress tolerance (**left**). The (**right**) panel summarize four major molecular pathways that modulate AS during stress: (**A**) splicing control via spliceosome dynamics mediated by SR proteins and hnRNPs; (**B**) hormone signaling exemplified by the ABA pathway, where ABA-bound PYR/PYL receptors inhibit PP2C phosphatases, activating SnRK2 kinases and downstream AS regulators; (**C**) RNA surveillance by NMD, which degrades aberrant isoforms while some isoforms are sequestered/stored; (**D**) epigenetic regulation in which chromatin-state changes exon–intron selection. Images were created with BioRender. Faiza Fatima. (2025). https://app.biorender.com/illustrations/68c05363ab47b2405e081788 (accessed on 12 September 2025).

**Figure 3 ijms-26-10146-f003:**
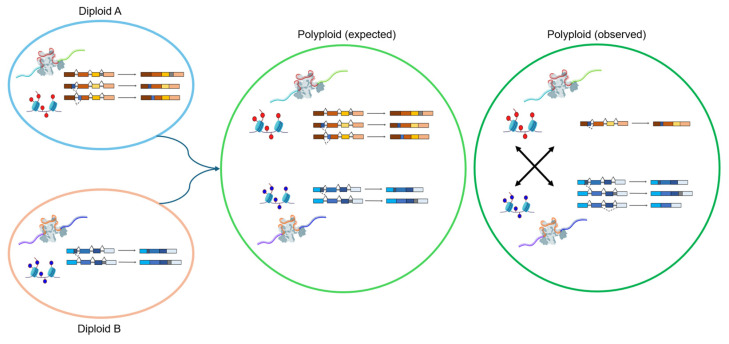
Conceptual model illustrating AS divergence between polyploid and its diploid progenitors. Diploid A and Diploid B represent parental lineages contributing to polyploid formation. Polyploid (expected) depicts theoretical additive expression and splicing outcomes assuming simple parental legacy. In contrast, polyploid (observed) reflects empirical findings across *B. napus*, *G. hirsutum*, and *T. aestivum*, where subgenome interactions, epigenetic remodeling, and *cis*/*trans* regulatory divergence reshape AS landscapes. These processes result in asymmetric subgenome activity, isoform switching, and altered AS isoform frequencies under stress and during developmental stages, leading to enhanced transcriptional plasticity and stress adaptation compared to diploid progenitors.

## Data Availability

No new data were created or analyzed in this study. Data sharing is not applicable to this article.

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
