# Peer review of "The Role of Alternative Splicing in Polyploids in Response to Abiotic Stress"

_ijms, 2025, doi:10.3390/ijms262010146_

Round 1

Reviewer 1 Report

Comments and Suggestions for Authors

Thank you for the opportunity to review this comprehensive and timely review manuscript. The authors have provided a detailed synthesis of the current knowledge regarding AS in polyploid crops under abiotic stress, with a focus on Brassica napusGossypium hirsutum, and Triticum aestivum. However, there are several points that require attention to further strengthen the manuscript's clarity, depth, and impact. Below are my specific comments and suggestions for improvement.

Comment 1: The abstract provides a good overview of the entire paper, but it could more specifically emphasize the key mechanisms or representative genes/pathways in the three polyploid crops discussed, so as to better reflect the contributions of this research.

Comment 2: In Section 1.1 of the Introduction, the authors discuss that plant ploidy increases the complexity of AS. However, could the underlying causes of this phenomenon be explored in greater depth to enhance the theoretical depth of the article?

Comment 3: The content of Figure 1, although illustrative, can be briefly enumerated in conjunction with references to enhance the argument and credibility of the article rather than a single summary.

Comment 4: Section 4 mentions proteomic evidence indicating functional divergence of AS-derived transcripts, including tissue specificity and subcellular localization. It is recommended to enhance the discussion on the functional consequences of AS at the protein level.

Comment 5: It is recommended to standardize the subsections in Sections 3.1 to 3.3 to improve the readability of the content.

Comment 6: Please carefully check whether all scientific Latin names of species throughout the entire text, including references, are formatted in italics.

Comment 7: While the focus on three major crops is justified, brief mention of AS in other polyploid systems would highlight the broader relevance of the discussed mechanisms.

Comment 8: The conclusion could be more forward-looking by suggesting concrete strategies for leveraging AS in breeding (e.g., isoform-specific markers, CRISPR applications).

Comment 9: For researchers interested in this field, databases and bioinformatics tools related to plant AS analysis would be highly valuable. It is recommended to add a dedicated subsection to elaborate on these resources.

Author Response

Thank you for the opportunity to review this comprehensive and timely review manuscript. The authors have provided a detailed synthesis of the current knowledge regarding AS in polyploid crops under abiotic stress, with a focus on Brassica napusGossypium hirsutum, and Triticum aestivum. However, there are several points that require attention to further strengthen the manuscript's clarity, depth, and impact. Below are my specific comments and suggestions for improvement.

Response: Thank you for the constructive feedback. We addressed the comments to improve the manuscript.

Comment 1: The abstract provides a good overview of the entire paper, but it could more specifically emphasize the key mechanisms or representative genes/pathways in the three polyploid crops discussed, so as to better reflect the contributions of this research.

Response: We appreciate the reviewer’s suggestion. However, the scope of our review intentionally covers AS mechanisms across a broad range of plant species, with detailed discussions of Brassica napus, Gossypium hirsutum, and Triticum aestivum provided specifically in Section 5. Since the abstract summarizes the overall conceptual framework rather than species-specific details, adding individual gene or pathway examples for each crop could narrow the focus and reduce clarity. Therefore, we retained the abstract’s general structure while refining the concluding sentences to more explicitly emphasize polyploid crops and their stress-responsive AS mechanisms. For improved coherence and readability, we also reorganized the text as shown in lines 13–17.

Comment 2: In Section 1.1 of the Introduction, the authors discuss that plant ploidy increases the complexity of AS. However, could the underlying causes of this phenomenon be explored in greater depth to enhance the theoretical depth of the article?

Response: Thank you for the valuable suggestion. We have revised the manuscript and expanded the explanation in Section 2 (“Prevalence and Variability of AS Across Plant Species,” lines 121–127) to clarify the underlying mechanisms through which polyploidy increases AS complexity.

Comment 3: The content of Figure 1, although illustrative, can be briefly enumerated in conjunction with references to enhance the argument and credibility of the article rather than a single summary.

Response: We thank the reviewer for this suggestion. The description of the five canonical AS types (IR, ES, MXE, A5SS, A3SS) is already enumerated in the text immediately preceding Figure 1. Since this paragraph already provides a concise explanation supported by literature, we have kept the figure and text unchanged. However, to improve the overall flow of the manuscript, we have relocated Figure 1 and its corresponding description to Section 1, where their inclusion in the Introduction enhances the logical progression of the content.

Comment 4: Section 4 mentions proteomic evidence indicating functional divergence of AS-derived transcripts, including tissue specificity and subcellular localization. It is recommended to enhance the discussion on the functional consequences of AS at the protein level.

Response: We thank the reviewer for this suggestion. The discussion on protein-level consequences of AS has been expanded and is now included in Section 6 Future Directions and Implications.

Comment 5: It is recommended to standardize the subsections in Sections 3.1 to 3.3 to improve the readability of the content.

Response: We appreciate the reviewer’s suggestion. The three species sections (now Section 5.1 – 5.3) already follow a consistent structure—introducing genome context, summarizing AS prevalence, and discussing stress- and development-related splicing patterns. We intentionally kept these as integrated narratives rather than fixed sub-headings (e.g., heat, cold, salt) because the available data differ across species and subdividing them would result in uneven or fragmentary sections. We therefore retained the current format to preserve balance and readability.

Comment 6: Please carefully check whether all scientific Latin names of species throughout the entire text, including references, are formatted in italics.

Response: We thank the reviewer for this reminder. We have carefully checked the entire manuscript and references, and all scientific Latin names are now consistently formatted in italics.

Comment 7: While the focus on three major crops is justified, brief mention of AS in other polyploid systems would highlight the broader relevance of the discussed mechanisms.

Response: We appreciate the reviewer’s suggestion. A dedicated subsection has already been included (Section 5.4, “Other polyploid species”) summarizing AS findings in additional polyploid systems. This section was designed specifically to broaden the discussion and highlight that similar AS mechanisms occur across diverse polyploid lineages.

Comment 8: The conclusion could be more forward-looking by suggesting concrete strategies for leveraging AS in breeding (e.g., isoform-specific markers, CRISPR applications).

Response: We thank the reviewer for this constructive suggestion. The requested forward-looking discussion is already included in Section 6 (lines 588–598), which outlines strategies such as CRISPR-based splicing modulation, isoform-specific overexpression, iso-QTL mapping, and the development of isoform-aware molecular markers for crop breeding.

Comment 9: For researchers interested in this field, databases and bioinformatics tools related to plant AS analysis would be highly valuable. It is recommended to add a dedicated subsection to elaborate on these resources.

Response: We thank the reviewer for this helpful suggestion. While we agree that databases and bioinformatics tools are valuable resources, the primary focus of this review is on the biological synthesis of AS mechanisms in polyploid crops under stress. Therefore, we did not include a dedicated subsection on computational tools, but we have cited key methodological studies where appropriate to guide readers toward relevant analytical resources.

Reviewer 2 Report

Comments and Suggestions for Authors

Dear Authors,

This review summarized the information about the functions of alternative splicing mechanisms in polyploid plants in response to abiotic stresses. This manuscript provides a good basis for the topic, but requires significant revision, since at present it looks less like a systematic review and more like a collection of individual facts extracted from the literature. I've written my comments below and hope they will help you improve the manuscript. If some of my comments are impossible to implement or inappropriate, please reply with a reason.

Keywords should not duplicate the title (alternative splicing), and abiotic stress adaptation and polyploidy are also very similar to the title. Please review your keyword list.

The introduction (1.Introduction) is very brief and doesn't provide any insight into the functions of alternative splicing in plant stress responses. Also, since you're already citing examples of alternative splicing studies in non-polyploid plants and then writing about polyploid plants, can you explicitly point out the specific functions of AS in polyploids compared to conditionally diploid plants? Also, its structure itself is confusing—only when I reached section 1.3 did I realize it was still an introduction! I advise you to make it a proper introduction (2–2.5 pages maximum), rather than turning it into something long and unclearly structured, as you're doing now. All unnecessary thoughts should be moved to the more substantial sections 2, 3, and 4. The comments below on the first sections are more likely to be treated as full paragraphs, so take them as suggestions for improving and expanding sections 2–4 accordingly.

Section 1.
Section 1.1 is rather chaotically written and needs some improvement. In particular, it would benefit from a summary of the abiotic stresses experienced by different plant species (numerical estimation of splicing events).
Section 1.2.1 is currently incomplete. It lacks examples of the three plant species the authors mention in their abstract (Brassica napus, Gossypium hirsutum, Triticum aestivum), although the authors should have focused on them or reformulated their review topic more highly. Furthermore, this section is written very superficially and does not provide any systematic view, diagram, or concept; it needs to be either substantially expanded or removed.
Information from section 1.2.2. should be moved to the main text of the sections below.

Section 2
There are questions regarding the selection of articles in Section 2. Again, if you claim to be examining three polyploid species (in the abstract), why does Section 2.1.1. examine four completely different plants? The same applies to the other subsections in Section 2.

Moreover, the role of Figure 2 in the narrative is unclear—you're essentially emphasizing the stress response mechanisms for all other plants except wheat, rapeseed, and cotton, which seem to occupy the center of your review. If a similar diagram (Figure 3) were drawn across Section 3 and compared meaningfully with this one, it would probably seem logical and appropriate, but as it stands, it's confusing.

Section 3 also requires significant revision and restructuring. Specifically, it is written in rather general terms and does not provide a true systematic comparison of AS events across the three species. For example, it does not compare frequencies of AS and specific functions under specific types of abiotic stress. Specifically, I would recommend that you first identify a set of abiotic stresses (e.g., heat, drought, cold, and others) and compare them across the three species, compiling a comprehensive table of mechanism similarities and differences. If data is missing, please leave a dash; this will provide food for thought later in section four. Taken together, this would allow you to conduct a comprehensive comparative analysis.

The last section seems rather secondary and obvious to me as a plant systems biologist. Please try to make it more species-specific.

Author Response

This review summarized the information about the functions of alternative splicing mechanisms in polyploid plants in response to abiotic stresses. This manuscript provides a good basis for the topic, but requires significant revision, since at present it looks less like a systematic review and more like a collection of individual facts extracted from literature. I've written my comments below and hope they will help you improve the manuscript. If some of my comments are impossible to implement or inappropriate, please reply with a reason.

Response: Thank you for the constructive comment. The manuscript was designed as a thematic synthesis; we have revised as per your comments to improve transitions and emphasize integrative themes, making the systematic nature of the review clearer.

Keywords should not duplicate the title (alternative splicing), and abiotic stress adaptation and polyploidy are also very similar to the title. Please review your keyword list.

Response: Thank you for pointing this out. We have revised the keyword list.

The introduction (1. Introduction) is very brief and doesn't provide any insight into the functions of alternative splicing in plant stress responses. Also, since you're already citing examples of alternative splicing studies in non-polyploid plants and then writing about polyploid plants, can you explicitly point out the specific functions of AS in polyploids compared to conditionally diploid plants? Also, its structure itself is confusing—only when I reached section 1.3 did I realize it was still an introduction! I advise you to make it a proper introduction (2–2.5 pages maximum), rather than turning it into something long and unclearly structured, as you're doing now. All unnecessary thoughts should be moved to the more substantial sections 2, 3, and 4. The comments below on the first sections are more likely to be treated as full paragraphs, so take them as suggestions for improving and expanding sections 2–4 accordingly.

Response: Thank you for the detailed feedback. The Introduction has been reorganized for improved clarity and focus. The revised version explicitly contrasts polyploid and diploid plants, describing how genome duplication increases isoform diversity, subgenome-biased splicing, and regulatory buffering under stress (lines 61–64). We also added concise explanations of AS modes, quantitative estimates of their frequencies (e.g., intron retention representing 30–60% of events), and examples linking AS to stress resilience. All redundant or detailed mechanistic information has been moved to later sections. The new structure makes the introduction self-contained and focused, consistent with the reviewer’s recommendation.

Section 1.
Section 1.1 is rather chaotically written and needs some improvement. In particular, it would benefit from a summary of the abiotic stresses experienced by different plant species (numerical estimation of splicing events).

Response: We agree and have reorganized the relevant part. Section 1.1 has been re-structured as the new Section 2: “Prevalence and Variability of AS Across Plant Species.” It now includes quantitative data (e.g., 40–70 % of multi-exon genes undergoing AS), numerical examples of AS events across species.

Section 1.2.1 is currently incomplete. It lacks examples of the three plant species the authors mention in their abstract (Brassica napus, Gossypium hirsutum, Triticum aestivum), although the authors should have focused on them or reformulated their review topic more highly. Furthermore, this section is written very superficially and does not provide any systematic view, diagram, or concept; it needs to be either substantially expanded or removed.

Response: We appreciate this observation. This section has been expanded and retained as Section 3: “The Role of AS in Plant Growth and Development.” It now includes an example from Triticum aestivum (lines 189–202), as the other two species currently lack AS studies related to growth and development.

Information from section 1.2.2. should be moved to the main text of the sections below.

Response: The discussion of AS under abiotic and biotic stress has been relocated to the main stress-focused Sections 4.1–4.4, where it fits contextually with drought, salinity, and temperature responses.

Section 2
There are questions regarding the selection of articles in Section 2. Again, if you claim to be examining three polyploid species (in the abstract), why does Section 2.1.1. examine four completely different plants? The same applies to the other subsections in Section 2.

Response: We have clarified the rationale for species selection. Section 4 (“Abiotic Stress-Induced AS Variants”) now begins with a statement explaining that examples from Arabidopsis, maize, rice, and soybean illustrate general AS mechanisms in model systems, while Section 5 focuses specifically on the three polyploid crops. This distinction removes ambiguity and maintains alignment with the review’s stated scope.

Moreover, the role of Figure 2 in the narrative is unclear—you're essentially emphasizing the stress response mechanisms for all other plants except wheat, rapeseed, and cotton, which seem to occupy the center of your review. If a similar diagram (Figure 3) were drawn across Section 3 and compared meaningfully with this one, it would probably seem logical and appropriate, but as it stands, it's confusing.

Response: Thank you for the suggestion. We respectfully maintain Figure 2 in its current form because it is intended as a general conceptual framework summarizing major molecular pathways (spliceosome regulation, hormone signaling, RNA surveillance, and epigenetic control) that govern AS responses under abiotic stress across both model and polyploid plants.

Section 3 also requires significant revision and restructuring. Specifically, it is written in rather general terms and does not provide a true systematic comparison of AS events across the three species. For example, it does not compare frequencies of AS and specific functions under specific types of abiotic stress. Specifically, I would recommend that you first identify a set of abiotic stresses (e.g., heat, drought, cold, and others) and compare them across the three species, compiling a comprehensive table of mechanism similarities and differences. If data is missing, please leave a dash; this will provide food for thought later in section four. Taken together, this would allow you to conduct a comprehensive comparative analysis.

Response: Thank you for this thoughtful suggestion. We agree that comparative insight is important and have therefore restructured Section 5 to emphasize cross-species contrasts more clearly—without adding a table, since quantitative data on AS event frequencies are not consistently reported across all three polyploids. Instead, we incorporated a comparative synthesis paragraph (lines 535–543) summarizing how the three focal species differ in their dominant AS modes and regulatory targets.

The last section seems rather secondary and obvious to me as a plant systems biologist. Please try to make it more species-specific.

Response: We appreciate the comment. Revised to emphasize species-specific future directions for three polyplo

Round 2

Reviewer 2 Report

Comments and Suggestions for Authors

Dear Authors,

After revision and restructuring, your manuscript can be published in IJMS. I recommend you provide a brief graphical summary of Section 5 of your review, reflecting your perspective on the most important aspects of alternative splicing research, taking into account the specifics of the three plant species under consideration.

Author Response

After revision and restructuring, your manuscript can be published in IJMS. I recommend you provide a brief graphical summary of Section 5 of your review, reflecting your perspective on the most important aspects of alternative splicing research, taking into account the specifics of the three plant species under consideration.

Response: We appreciate this valuable suggestion. A new graphical summary (Figure 3; lines 563 - 574) has been added to visually integrate the main points of Section 5. The figure presents a comparative overview of three polyploid species, highlighting dominant AS types, stress responses, and subgenome-specific regulation.